# Cost-effectiveness of pulse oximetry and integrated management of childhood illness for diagnosing severe pneumonia

**Solomon H. Tesfaye**[1,2,3]*, **Eskindir Loha**[1,2], **Kjell Arne Johansson**[2], **Bernt Lindtjørn**[1,2]

**1** School of Public Health, Hawassa University, Hawassa, Ethiopia, **2** Centre for International Health, University of Bergen, Bergen, Norway, **3** School of Public Health, Dilla University, Dilla, Ethiopia

* solomon0917242124@gmail.com

**Data Availability Statement:** All relevant data are within the paper.

**Funding:** The authors received no specific funding for this work.

## Abstract

Pneumonia is a major killer of children younger than five years old. In resource constrained health facilities, the capacity to diagnose severe pneumonia is low. Therefore, it is important to identify technologies that improve the diagnosis of severe pneumonia at the lowest incremental cost. The objective of this study was to conduct a health economic evaluation of standard integrated management of childhood illnesses (IMCI) guideline alone and combined use of standard IMCI guideline and pulse oximetry in diagnosing childhood pneumonia. This is a cluster-randomized controlled trial conducted in health centres in southern Ethiopia. Two methods of diagnosing pneumonia in children younger than five years old at 24 health centres are analysed. In the intervention arm, combined use of the pulse oximetry and standard IMCI guideline was used. In the control arm, the standard IMCI guideline alone was used. The primary outcome was cases of diagnosed severe pneumonia. Provider and patient costs were collected. A probabilistic decision tree was used in analysis of primary trial data to get incremental cost per case of diagnosed severe pneumonia. The proportion of children diagnosed with severe pneumonia was 148/928 (16.0%) in the intervention arm and 34/876 (4.0%) in the control arm. The average cost per diagnosed severe pneumonia case was USD 25.74 for combined use of pulse oximetry and standard IMCI guideline and USD 17.98 for standard IMCI guideline alone. The incremental cost of combined use of IMCI and pulse oximetry was USD 29 per extra diagnosed severe pneumonia case compared to standard IMCI guideline alone. Adding pulse oximetry to the diagnostic toolkit in the standard IMCI guideline could detect and treat one more child with severe pneumonia for an additional investment of USD 29. Better diagnostic tools for lower respiratory infections are important in resource-constrained settings, especially now during the COVID-19 pandemic.

## Introduction

Pneumonia is a leading cause of death among children younger than five years old [1]. Childhood pneumonia is associated with chronic obstructive pulmonary disease, reduced lung

**Competing interests:** The authors have declared that no competing interests exist.

function, and chronic bronchitis during adulthood [2]. The World Health Organization's integrated management of childhood illness (IMCI) guideline has had a huge impact on clinical management of severe pneumonia in children aged 2 months to 59 months in resource constrained settings [3]. However, the diagnostic recommendations in this guideline are not sensitive enough to detect or specific enough to rule out severe pneumonia [4, 5]. Thus, even though the standard IMCI guideline improves the identification and treatment of pneumonia [6], severe cases with high risk of mortality still may be missed. Combining pulse oximetry with the standard IMCI guideline may improve diagnostic precision and thus prevent childhood deaths due to severe pneumonia [7].

Implementation of the standard IMCI guideline requires training staff, provision of essential drugs and supplies, and consistent supervision [8]. These requirements can be challenging in low-income countries [9, 10]. Implementation of the standard IMCI guideline in Ethiopia can be difficult due to a lack of trained staff, a lack of essential drugs and supplies, and inconsistent supervision [9, 10]. For example, Ethiopia officially adopted the IMCI guideline in 1997 [8], but around 30,700 children under five years old continue to die due to pneumonia annually in the country [11]. More evidence on how to improve the effectiveness of standard interventions and on the cost of adding advanced diagnostics tools to existing pneumonia policies is needed in Ethiopia and other low-income countries.

Administering pulse oximetry and oxygen therapy improves hospital management of pneumonia [12, 13]. In Papua New Guinea, improving oxygen system and pulse oximetry has improved the quality of care and reduced mortality from pneumonia by 35% among children age younger than 5 years [14]. Incorporation of pulse oximetry into the usual clinical management of pneumonia among young children in developing countries thus seems promising [15–17]. From our previous cluster-randomized controlled trial, we found that combining pulse oximetry with the standard IMCI guideline improves the capacity of health workers: diagnoses of severe childhood pneumonia increased from 4% using the standard IMCI guideline alone to 16% using the combination of pulse oximetry with the standard IMCI guideline [18].

Several studies have assessed the effect of adding pulse oximetry in settings where the standard IMCI guideline already is used to manage childhood pneumonia [17, 19–21]. However, none of those studies estimated the opportunity cost of adding pulse oximetry to standard IMCI guideline. Policy makers use cost-effectiveness analysis to identify economical and effective purchases, and this analysis is especially important in health care settings [22]. Health economic evaluations based on randomized controlled trials in policy decision making may prevent countries with scarce resources not to waste health investments [23]. This study therefore aimed to compare the cost-effectiveness of combined use of pulse oximetry with standard IMCI guideline compared with the standard IMCI guidelines alone to improve diagnostic precision of severe childhood pneumonia in rural Ethiopia.

## Materials and methods

### Ethics statement

The study was approved by the institutional review board of the College of Medicine and Health Sciences at Hawassa University (ref: IRB/009//2017) and the Regional Committees for Medical Research Ethics, South East Norway (ref: 2017/2473/REK sør-øst). Children were included in the study after obtaining written informed consent from parents.

### Study design and settings

This cost-effectiveness study was conducted alongside a cluster-randomized controlled trial whose main objective was to improve diagnosis of severe childhood pneumonia by adding

pulse oximetry to the standard IMCI guideline. A detailed description of the trial methodology and setting is provided elsewhere [18]. The trial is registered with trial registration number: PACTR, PACTR201807164196402 in 14/06/2018 and available at https://pactr.samrc.ac.za/TrialDisplay.aspx?TrialID=3466. Briefly, 24 government health centres were studied. Each health centre was defined as a cluster, and clusters with at least one case of pneumonia per day were included in the study. Children aged two months to 59 months who sought care at a health centre for cough or difficulties breathing lasting fewer than 14 days were eligible for inclusion in the study. Recruitment took place between September 2018 and April 2019.

The study was conducted in *Gedeo* district located in Southern part of Ethiopia. Out of a total population of more than one million, about 173,000 are children under five years of age. The district has 38 health centres, three of which were recently upgraded to primary hospitals. It has 146 health posts and one teaching and referral hospital. IMCI is implemented in health centres for the management of common childhood illnesses, including pneumonia. Radiology or laboratory diagnostic tools are neither available nor required to diagnose childhood pneumonia in such settings. Health centres are expected to refer severe pneumonia cases to a primary hospital after offering the recommended pre-referral drugs. None of the health centres had pulse oximetry for measuring oxygen saturation. Oxygen therapy for severely ill children is available at one hospital, but the supply is unreliable. In 2018, 106,583 outpatient visits were reported in the 24 health centres selected for the trial, representing 85% of all outpatient visits in the study area. Of those, 22,542 (21%) visits were made by children younger than five, and 6,677 (30%) of those children were diagnosed with pneumonia.

## Description of the interventions compared

The detailed description of the interventions and the sample size of the population are provided in the published trial article [18]. Twelve clusters were randomly selected into each of the two arms. Health workers in the intervention arm used the standard IMCI guideline [24], and a paediatric fingertip pulse oximetry (ADC Adimals 2150) to diagnose pneumonia. The health workers measured each child's oxygen saturation by taking two pulse oximetry measurements at five minutes apart. One-day training was given to health workers on the use of pulse oximetry. Health workers in the control arm used the same IMCI guideline alone to manage children with suspected signs and symptoms of pneumonia. All health workers were trained on the standard IMCI guideline.

## Measurement of health effects

The health effect for this trial based cost-effectiveness analysis was based on cluster randomised trial [18]. In the intervention group, the measure of effectiveness was severe pneumonia cases detected using the standard IMCI guideline [24] with or without hypoxemia (oxygen saturation < 90%), as measured by a paediatric fingertip pulse oximetry (ADC® Adimals 2150). In the control group, severe pneumonia was detected using the same IMCI guideline alone. Because of the lack of reference standard diagnostic method in the study settings we didn't estimate the improved health outcome and the averted cost related to improved diagnosis at the study settings. We use detected severe pneumonia cases using pulse oximetry and IMCI guidelines as health outcome for the cost-effectiveness analysis. Using such intermediate outcome is also recommended by Drummond et al. [23].

## Intervention costs measurement

Intervention costs of diagnosing severe pneumonia were assessed from both the provider and patient perspectives [23], using 2018 US dollars. We use the word "provider" to refer to the

health systems/institutions. All costs were converted to US dollars using the official National Bank of Ethiopia average exchange rate for 2018 (US dollar 1 = Ethiopian Birr 27.4220). Data were collected prospectively, starting from the beginning of the trial.

We used a structured questionnaire to obtain cost information. The type, quantity, and price of each resource used in the trial were recorded via interviews with caregivers and by assessing facility records. Patient costs included patient's direct out-of-pocket expenses for consultations, transportation, drugs, supplies (Intravenous fluids and Intravenous cannula 24 gages), and hospital admission. Patients pay out-of-pocket for drugs and supplies from government health facilities, and the government applies a 25% surcharge to these items. Costs for drugs and supplies represent both the health system costs and patient costs. Therefore, we include cost for drugs and supplies on patient side to avoid double counting. For this cost-effectiveness analysis we considered drugs and supplies, hospital stays and transportation costs as patient opportunity cost. Drug costs include cost for single dose of antibiotics given at health centres level before referral to hospital. Cost for drugs and supplies, hospital stays, oxygen therapy and IV fluids were included at hospitals for severe pneumonia cases admitted to hospital. The unit costs for all resources used for diagnosing pneumonia, oxygen treatment, training, and patient expenses are presented in Table 1.

Provider cost items were divided into capital and recurrent costs. Capital costs were defined as costs for items expected to last longer than a year [23], such as a pulse oximetry. The capital cost of the pulse oximetry was recorded from invoices and annuitized based on an expected product life of two years [25], initial costs, and interest rate of 7% [23]. Recurrent costs were defined as costs for products used regularly that have duration of less than a year. We included costs of personnel (including health workers' time spent diagnosing childhood pneumonia), oxygen therapy (provided by the government), training of health workers (including materials and estimated based on the per diem used by the district health office), and pulse oximetry alkaline batteries. To estimate the average cost of health workers' time spent on diagnosis of pneumonia, a monetary value was assigned by allocating a corresponding percentage of the health worker's salary and duty fees (this is payment for extra working hours). The salary per year in the study area for mid-level health worker is USD 1, 474, and a duty fee is USD 2,580. All the mid-level health workers in the study area are Bachelor of Science degree holder with four years medical education background and their salary is the same.

**Table 1. Unit cost of items for pneumonia diagnosis and treatment, 2018 USD.**

| Items | Unit costs | |
|---|---|---|
| Diagnosis | Intervention arm | Control arm |
| Pulse oximetry | 149.81 | Not applicable |
| Batteries | 1.01 | Not applicable |
| Personnel | 0.59 | 0.34 |
| Training | 13.96 | 7.40 |
| Oxygen treatment (per cubic meter) | 0.77 | 0.77 |
| Patient expense | | |
| Drugs | 1.01 | 1.01 |
| Intravenous fluids | 0.88 | 0.88 |
| Intravenous cannula 24 gages | 0.34 | 0.34 |
| Hospital stay | 1.18 | 1.18 |
| Consultation | 0.40 | 0.40 |
| Transportation per kilometre | 0.03 | 0.03 |

The cost of drugs, intravenous fluids, and intravenous cannula 24 gages used were quantified and multiplied by their respective unit costs to estimate the total cost of each items in both arms. The total cost of hospital stays was also estimated by multiplying the cost per day by the duration of hospital stays. Similarly, we estimated the total cost of oxygen therapy by multiplying the amount of oxygen consumed in cubic meter by the unit cost. The total distance travelled by the patient from home to health facilities was multiplied by the unit cost per kilometre to estimate the total transportation costs. All costs for each items was added up to get total costs for each arms. Total and average costs per diagnosed severe and non-severe pneumonia cases were estimated. To estimate the average cost per diagnosed severe pneumonia case, the total cost was divided by the number of children diagnosed with severe pneumonia. Likewise the total cost was divided by the number of children diagnosed for non-severe pneumonia.

## Cost-effectiveness model

A decision tree model built using TreeAge Pro Suit 2021 (© 2021 TreeAge software, Inc.) was used for the analysis [26]. The model follows a sequence of steps to construct a tree structure under uncertainty for alternative interventions and select the least expected cost per benefit as the best alternative. A cost-effectiveness ratio was estimated for each of diagnostic methods per diagnosed severe pneumonia case. The cost-effectiveness model is presented in Fig 1. The model compares the opportunity cost and the proportion of severe pneumonia cases detected using combined use of pulse oximetry and the standard IMCI guideline versus the standard IMCI guideline in an Ethiopian rural setting. Probabilities in the decision tree represent possible events in the process of diagnosis after a child presents with symptoms of lower respiratory infection. The pathways are mutually exclusive. We conducted the study in rural part of Ethiopia where non-clinicians are responsible for child health care. Once a child is identified with severe pneumonia the health workers should refer the child to hospital where clinicians are responsible for child care. As a result we didn't include the final health outcome at hospital

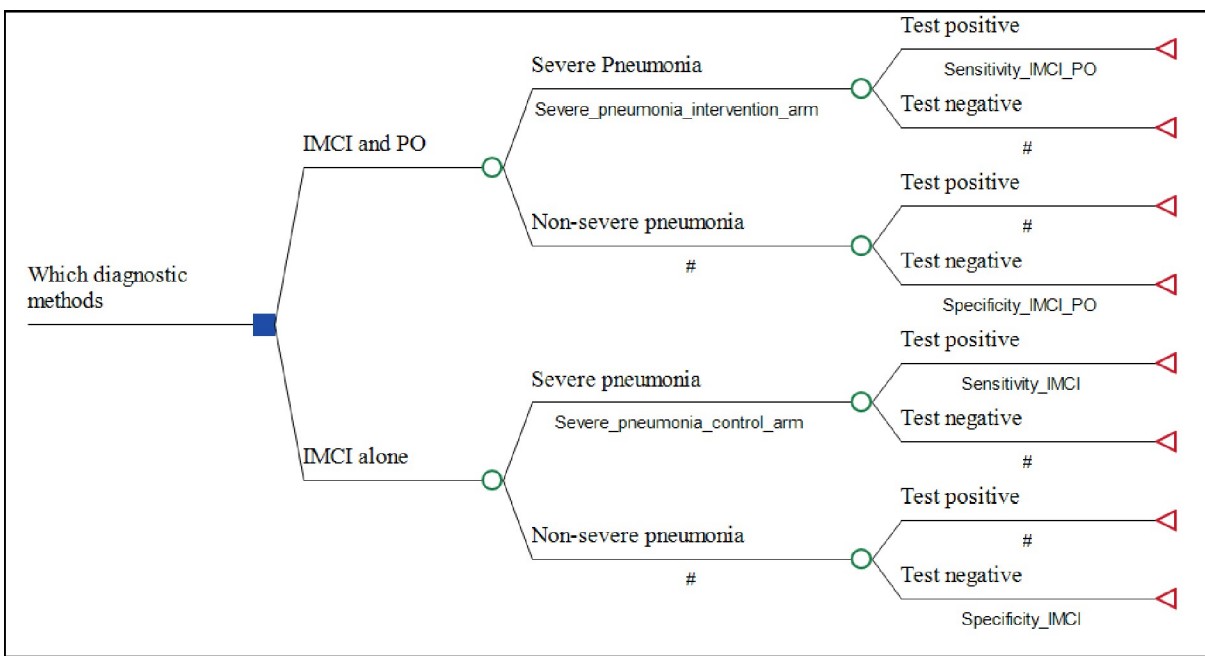

**Fig 1. Model structure of the cost-effectiveness study.**

level in the cost-effectiveness model. Therefore, in this study we didn't analyse improved health outcomes and averted costs to the health system and patient, averted medical costs, averted health system expenditures from improved diagnosis at health centre levels. To use mortality averted or life years gained due to interventions as final end point is not feasible, due to the fact that we conducted the study for short period of time (8 moths). During this period there were only four deaths.

Table 2 shows the input parameters, and specifies sources of those data. To estimate the proportion of diagnosed severe pneumonia cases for the two diagnostic modalities, we divided the number of severe pneumonia cases identified in the study period by the total number of children that attended the health centers. We make an assumption about the ability of both diagnostic modalities to accurately classify a child as severe pneumonia (sensitivity) and non-severe pneumonia (specificity). We used the sensitivity and specificity for both diagnostic modalities from literatures. Although data were available on the sensitivity and specificity of the standard IMCI guideline, sufficient data were not available to inform the specificity of standard IMCI guideline when combined with pulse oximetry. Therefore, we assumed the specificity of pulse oximetry combined with standard IMCI guideline to be similar with specificity of the standard IMCI guideline.

## Cost-effectiveness analysis

Incremental cost-effectiveness ratio (ICER) was used to summarize and present the cost-effectiveness result computed for the number of severe pneumonia cases identified as a result of the intervention.

## Uncertainty and sensitivity analysis

Two types of sensitivity analyses were performed to deal with uncertainties. First, a one-way sensitivity analysis was done using a tornado diagram for maximum and minimum values of costs, proportion of diagnosed severe pneumonia cases, and sensitivity and specificity of the interventions from the base case. The minimum and maximum values of the 95% confidence interval were used for both outcome and cost (Table 2). Second, probabilistic sensitivity analysis (PSA) was conducted to distribute the parameters used for one-way sensitivity analysis. PSA was performed with Monte Carlo simulations with 10,000 iterations. We assumed cost

**Table 2. Input parameters for cost-effectiveness model.**

| Input parameters | Base value | Minimum value | Maximum value | SD | Distribution | Data sources |
|---|---|---|---|---|---|---|
| Cost of diagnosed severe pneumonia with pulse oximetry and IMCI combined | 25.74 | 14.87 | 34.51 | 3.62 | Gamma | Trial data [18] |
| Cost of diagnosed non severe pneumonia with pulse oximetry and IMCI combined | 3.58 | 2.84 | 4.36 | 0.34 | Gamma | Trial data [18] |
| Cost of diagnosed severe pneumonia with IMCI | 17.98 | 14.23 | 24.55 | 2.97 | Gamma | Trial data [18] |
| Cost of diagnosed non-severe pneumonia with IMCI | 2.14 | 1.46 | 2.97 | 0.33 | Gamma | Trial data [18] |
| Proportion of severe pneumonia with pulse oximetry and IMCI combined | 0.16 | 0.05 | 0.27 | 0.03 | Beta | Trial data [18] |
| Proportion of severe pneumonia with IMCI | 0.04 | 0.01 | 0.07 | 0.01 | Beta | Trial data [18] |
| Sensitivity of pulse oximetry and IMCI combination | 0.85 | 0.72 | 0.98 | 0.07 | Beta | [27] |
| Specificity of pulse oximetry and IMCI combination | 0.87 | 0.73 | 1.00 | 0.07 | Beta | Assumed to be similar to IMCI |
| Sensitivity of IMCI | 0.56 | 0.39 | 0.73 | 0.09 | Beta | [28] |
| Specificity of IMCI | 0.87 | 0.73 | 1.00 | 0.07 | Beta | [29] |

parameters to follow gamma distributions and proportions of outcome and sensitivity and specificity of interventions to follow beta distributions [30]. In the analysis, we replaced the variables in the model with distributions. The results are presented as cost-effectiveness acceptability curves and scatter plots. The time horizon for cost-evaluation was 8 months, as this is the data collection period and average cost was calculated by dividing total cost by all diagnosed cases over the study period. We used discount rate of 0% both for cost and outcome as recommended for economic evaluation with short time horizon (< 1 year) [31].

## Results

### Baseline characteristics

The flow chart and baseline characteristics of the study participants are presented elsewhere [18]. We included 1,804 participants (928 in the intervention group and 876 in the control arm) from 24 health centres.

### Effectiveness

The proportion of children diagnosed with severe pneumonia was 148/928 (16.0%, 95% CI 4.7–27.2) in the intervention arm and 34/876 (4.0%, 95% CI 1.2–6.6) in the control arm.

### Cost of interventions

The total cost was USD 3,809.24 for severe pneumonia and USD 2,794.45 for non-severe pneumonia in the intervention arm. The total cost was USD 611.30 for severe pneumonia and USD 1,800.17 for non-severe pneumonia in the control arm. The cost per diagnosed severe pneumonia is USD 25.74 for intervention and USD 17.98 for control arms (Table 3). Of the total costs, the cost for drugs and supplies, transportation, and hospital stay account for most of the costs in both the intervention and control arms (Table 4).

### Cost-effectiveness

The ICER from base case analysis was USD 29 for standard IMCI guideline and pulse oximetry combined for diagnosing one additional severe pneumonia case as compared to standard IMCI guideline alone.

### Sensitivity analysis

One-way sensitivity analysis with minimum and maximum values of the selected variables is presented in Fig 2. The tornado diagram indicates that the cost of diagnosing severe pneumonia using standard IMCI guideline alone and sensitivity of combined use of standard IMCI guideline and pulse oximetry had the highest impact on the incremental cost-effectiveness ratio. The ICER ranged from USD 26.24 to USD 140.35 when the cost of diagnosing severe

**Table 3. Total and average cost for diagnosed pneumonia cases, 2018 USD.**

| Diagnostic alternatives | Severe pneumonia | | | non-severe pneumonia | | |
|---|---|---|---|---|---|---|
| | Number diagnosed | Total cost (USD) | Average cost (USD) | Number diagnosed | Total cost (USD) | Average cost (USD) |
| Standard IMCI alone | 34 | 611.30 | 17.98 | 842 | 1800.17 | 2.14 |
| Standard IMCI and pulse oximetry combined | 148 | 3809.24 | 25.74 | 780 | 2794.45 | 3.58 |

**Table 4. Itemized cost to diagnose severe pneumonia, 2018 USD.**

| Costs | Pulse oximetry with integrated management of childhood illnesses (% share) | Integrated management of childhood illnesses alone (% share) |
|---|---|---|
| **Provider costs** | | |
| Pulse oximetry | 191 (5) | Not applicable |
| Batteries | 1 (0.01) | Not applicable |
| Training | 27 (1) | 3 (1) |
| Personnel | 135 (4) | 15 (2) |
| Oxygen therapy | 743 (20) | 171 (28) |
| **Patient costs** | | |
| Consultation | 59 (2) | 14 (2) |
| Drugs and supplies | 1221 (32) | 181 (30) |
| Transportation | 559 (15) | 139 (23) |
| Hospital stay | 873 (23) | 89 (14) |
| Total cost | 3809 (100) | 611 (100) |

pneumonia using standard IMCI guideline alone varied from USD 14.23 to USD 24.55. The ICER was less sensitive to change in most of the other variables.

Figs 3 and 4 show the probabilistic sensitivity analysis results using the cost-effectiveness analysis scatterplot and cost-effectiveness acceptability curve. Fig 3 indicates that there was less variability both in cost and effectiveness of the standard IMCI guideline alone, but variability in standard IMCI and pulse oximetry was high.

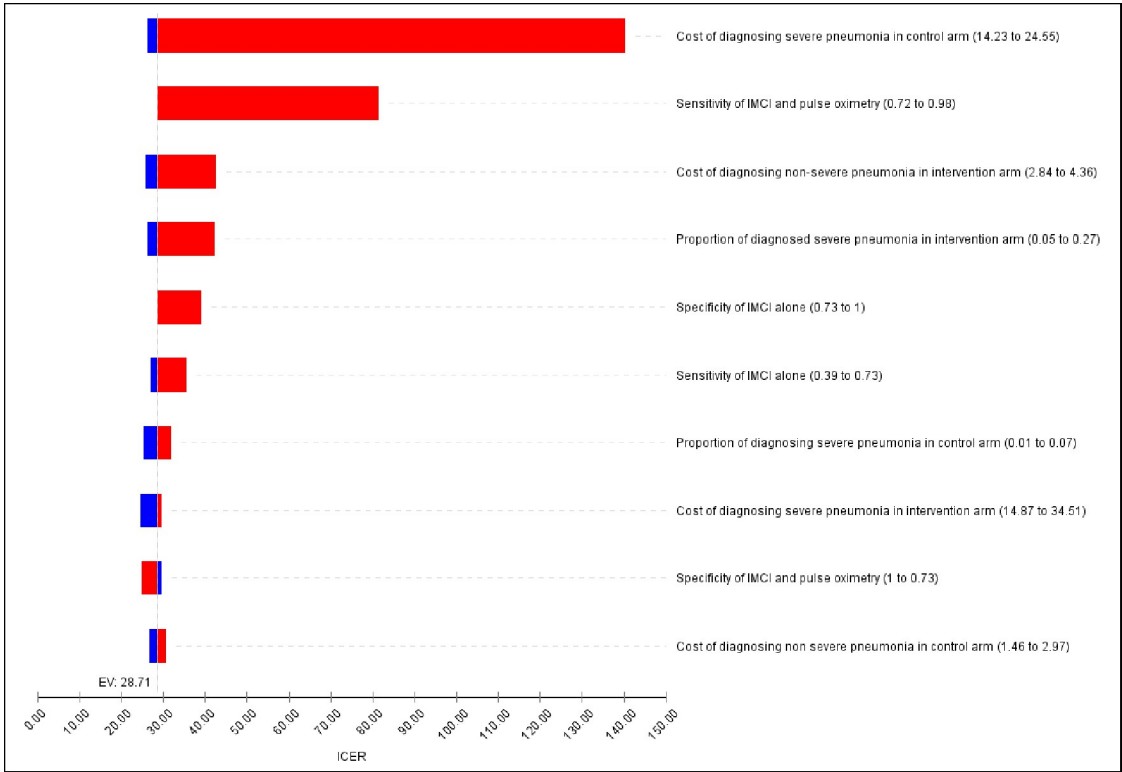

**Fig 2. Tornado diagram-sensitivity of ICER variations.**

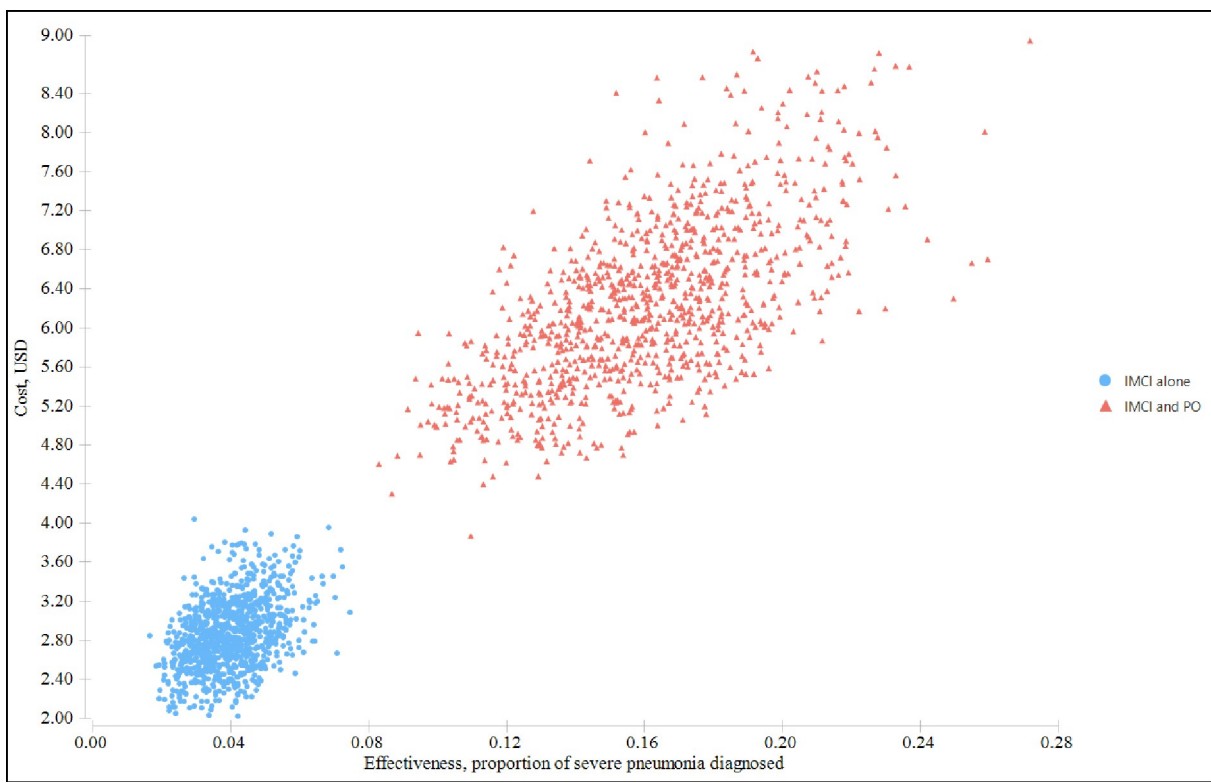

**Fig 3. Scatterplot of the costs and health effects of interventions from the Monte Carlo simulation.**

In a probabilistic sensitivity analysis, the ICER ranges from USD 7.16 to USD 89.29 per diagnosed severe pneumonia case. The acceptability curve in Fig 4 plots the proportion of iterations in which each alternative had the greater ICER for different Willingness-to-pay thresholds. The probability of combined use of standard IMCI guideline with pulse oximetry being cost-effective was 33% at a willingness-to-pay threshold of USD 26 per diagnosed severe pneumonia case. While at willingness-to-pay threshold of USD 39 per severe pneumonia case diagnosed, the rank order swops and the probability of combined use of standard IMCI with pulse oximetry appears with the highest probability of being cost-effective.

## Discussion

We found that pulse oximetry in combination with the standard IMCI guideline approach increased the detection of severe pneumonia in children by 12%. The ICER was 29 USD per severe case of pneumonia detected, compared with using the standard IMCI guideline alone. The results show that one more child with severe pneumonia could be detected and treated for an additional investment of USD 29. The combination of the standard IMCI guideline with pulse oximetry is more likely to be cost-effective, compared to IMCI guideline alone, at willingness-to-pay threshold of USD 39 per severe pneumonia case diagnosed. These findings should be useful for policy makers in defining the benefit package for management of lower respiratory infections in resource constrained settings.

A direct comparison of our findings with other cost-effectiveness analyses is difficult due to differences in study design, costing perspectives, and effectiveness of measurement. Nevertheless, some findings from other studies are comparable to ours. For example, providing oxygen system and pulse oximetry in Papua New Guinea, found that the ICER was USD 51 per treated

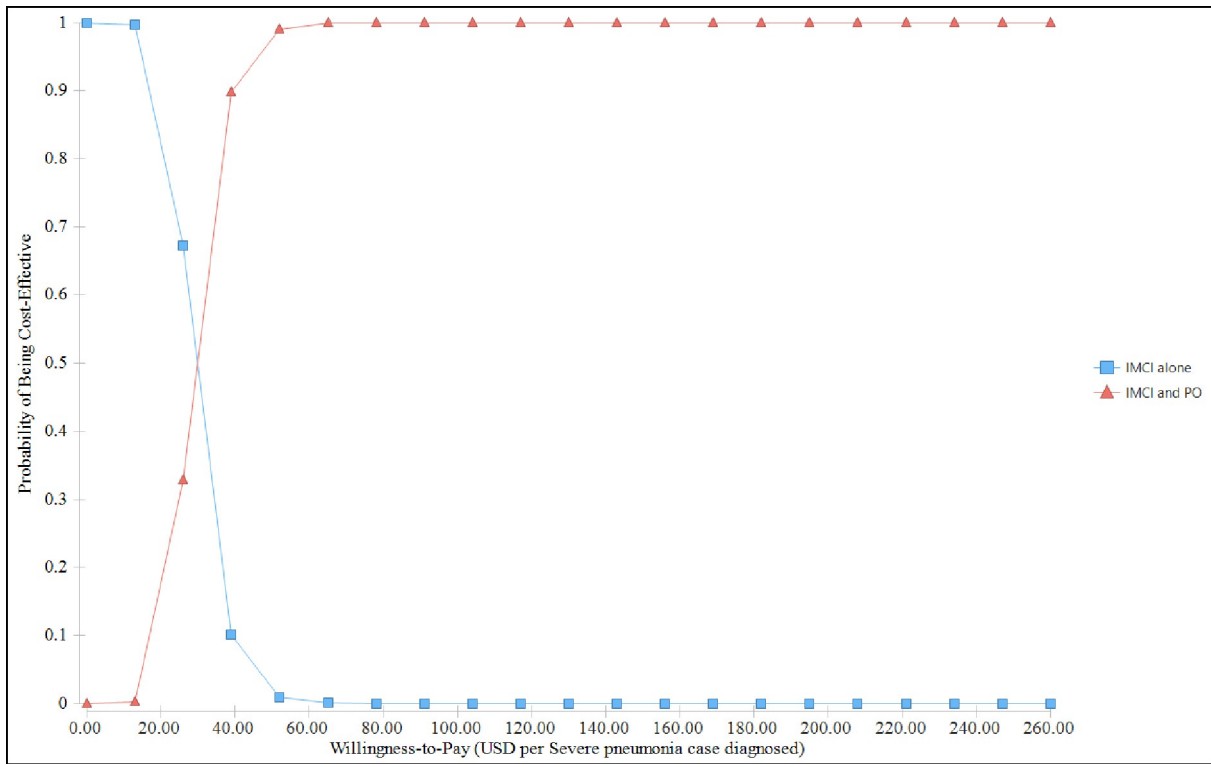

**Fig 4. Cost-effectiveness acceptability curve.**

severe pneumonia case [14]. A modelling study of 15 countries with high pneumonia mortality rates showed that in Ethiopia, combined use of pulse oximetry and the standard IMCI had an ICER of USD 6 per disability-adjusted life-years averted [32]. However, this modelling study collected cost data only from the provider perspective and excluded costs related to oxygen therapy, parenteral antibiotics, and hospital stays.

Data from 74 countries, including Ethiopia, were used to model the cost-effectiveness of the IMCI guideline [33]. In their study, the IMCI guideline has a median cost-effectiveness ratio of USD 26.6 (interquartile range: 17.7–45.9) per disability-adjusted life-years averted. Another study from Zambia revealed that the cost per outpatient visit for pneumonia management using the IMCI guideline is USD 48 per out-patient visit [34]. We found that the average cost for diagnosis of pneumonia using the IMCI guideline alone was lower than that found in both of these studies. This difference could be due to the inclusion of health facility building mainte-nance cost included in these two studies.

Our trial revealed that direct out-of-pocket payments for drugs, supplies, transportation, and hospital admission in both the intervention and control arms are the main cost compo-nents. Private expenditures pose a substantial financial risk to households and could be an important barrier for seeking health care [35, 36]. A study from Ethiopia showed that among total household out-of-pocket expenditures, costs of medication, hospital stays, and diagnostic investigations are the most important private health expenditures [36]. Moreover, 7% of households in Ethiopia with severe pneumonia fall below the extreme poverty line due to out-of-pocket payments to health care [36].

Universal health coverage is a sustainable development goal to be achieved by 2030 [37]. To achieve this goal, countries must develop health financing systems so that people can access

services without incurring financial hardship [38]. Accordingly, the Ethiopian Federal Ministry of Health has endorsed a health care financing strategy [39]. A fee-wavier system for the poor is one of the reforms included in this strategy. However, as shown in our trial and others [36], the largest out-of-pocket payments are for drugs, supplies, transportation, and hospital stays. Ethiopia therefore may be far from achieving universal health coverage.

We found that the main drivers of increased costs in the intervention arm were the capital cost of the pulse oximetry unit and the recurrent costs after diagnosis of pneumonia, such as medication, transportation, and hospital stays. However, the societal gain in terms of health benefit from using pulse oximetry is huge. Comparing this health benefit from using pulse oximetry against the consequences of false positive severe pneumonia identified using pulse oximetry should be the focus of future research area.

The ICER from the probabilistic sensitivity analysis doesn't change the conclusion from the deterministic analysis result, which implies low uncertainty.

The strength of this study is that data were collected prospectively in a cluster-randomized controlled trial. Resource identification and costing information were compiled based on routine care for childhood pneumonia at health centres in Ethiopia. The information was collected from typical rural health facilities in Ethiopia. Cost data were collected prospectively to reduce recall bias. Randomization was performed at the cluster level to avoid contamination. Broad-range cost categories were included, and the analysis accounted for the hierarchical nature of the data.

This cost-effectiveness analysis is not without limitations. First, we did not use a final endpoint, such as life years gained, as a health outcome. Instead, we used severe pneumonia diagnosis, which is an intermediate outcome. In economic evaluations of clinical trials, using intermediate outcome is misleading unless there is an established link between intermediate and final outcomes [23]. However, it is well-established that severe pneumonia is a leading cause of death in children under five years old [15, 40]. Additionally, it is reasonable to use diagnosed severe pneumonia as a health outcome if there is a clinical and cost-effective therapy to treat detected cases [23]. Second, our estimation did not include caregivers' lost productivity due to time spent seeking care and caring for a sick child. This omission might have led to underestimating the cost-effectiveness ratio. Third, the capital building costs was not included in the cost estimates. However, we expect that improved diagnostics with pulse oximetry may not have a huge impact on capital facility costs compared to the standard intervention assessed in the trial. Therefore it is unlikely, the exclusion of building cost to have substantial impact on ICER.

The study area is typical of the rural population of Ethiopia, where outpatient visits due to pneumonia are high and pulse oximetry is not available. Ethiopia's Federal Ministry of Health plans to ensure regular availability and functionality of oxygen therapy and pulse oximetry [41], although this has not yet been achieved. Therefore, we believe that our findings can be applied in rural health centres where diagnostic capacity is low. They also can help the ongoing efforts by the Federal Ministry of Health of Ethiopia to improve health care access in rural areas.

## Conclusions

Based on our trial findings, supplementing pulse oximetry with the standard IMCI guideline resulted in higher detection of severe childhood pneumonia than using the standard IMCI guideline alone. Therefore, using a combination of pulse oximetry and the standard IMCI guideline has both economic and public health importance.

## Acknowledgments

We sincerely acknowledge the contributions of the Gedeo Zone Health Department and district health offices in helping to successfully launch the implementation of this study. We are grateful for the health workers and health facilities where the study was conducted. We also sincerely thank the study participants.

## Author Contributions

**Conceptualization:** Solomon H. Tesfaye, Bernt Lindtjørn.

**Data curation:** Solomon H. Tesfaye.

**Formal analysis:** Solomon H. Tesfaye, Kjell Arne Johansson.

**Funding acquisition:** Bernt Lindtjørn.

**Investigation:** Solomon H. Tesfaye, Bernt Lindtjørn.

**Methodology:** Solomon H. Tesfaye, Kjell Arne Johansson, Bernt Lindtjørn.

**Resources:** Bernt Lindtjørn.

**Software:** Solomon H. Tesfaye.

**Supervision:** Eskindir Loha, Kjell Arne Johansson, Bernt Lindtjørn.

**Writing – original draft:** Solomon H. Tesfaye.

**Writing – review & editing:** Solomon H. Tesfaye, Eskindir Loha, Kjell Arne Johansson, Bernt Lindtjørn.

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
