## [Decision Letter · Decision Letter 0]

29 Mar 2022

PGPH-D-22-00195

Cost-effectiveness of Pulse oximetry and integrated management of childhood illness for diagnosing severe Pneumonia

Dear Dr. Tesfaye,

Thank you for submitting your manuscript to PLOS Global Public Health. After careful consideration, we feel that it has merit but does not fully meet PLOS Global Public Health’s publication criteria as it currently stands. Therefore, we invite you to submit a revised version of the manuscript that addresses the points raised during the review process.

We invite you to submit a revised version of the above manuscript, taking into consideration the comments received from the two reviewers. In particular, please address the second reviewer's detailed comments regarding the methods and consider adding a table to show how the input costs were derived. In addition, please be sure to address the first reviewer's concern about facility space, i.e., by either incorporating this cost category in the cost estimates or clearly explaining the rationale for and likely impact of omitting this cost category. Finally, please carefully check grammar and ensure all issues have been corrected. For example, common nouns (e.g., pulse oximetry, pneumonia) should not be capitalized unless they come at the beginning of a sentence.

We look forward to receiving your revised manuscript.

Kind regards,

Melissa Morgan Medvedev, M.D., Ph.D.

Academic Editor

Journal Requirements:

1. Please ensure you have included the registration number for the clinical trial referenced in the manuscript.

2. Please amend your Financial Disclosure statement. If you did not receive any funding for this study, please simply state: “The authors received no specific funding for this work.”

3. Please provide separate figure files in .tif or .eps format only and ensure that all files are under our size limit of 20MB.

Additional Editor Comments (if provided):

Reviewers' comments:

Reviewer's Responses to Questions

**Comments to the Author**

1. Does this manuscript meet PLOS Global Public Health’s publication criteria? Is the manuscript technically sound, and do the data support the conclusions? The manuscript must describe methodologically and ethically rigorous research with conclusions that are appropriately drawn based on the data presented.

Reviewer #1: Yes

Reviewer #2: Partly

2. Has the statistical analysis been performed appropriately and rigorously?

Reviewer #1: Yes

Reviewer #2: N/A

3. Have the authors made all data underlying the findings in their manuscript fully available (please refer to the Data Availability Statement at the start of the manuscript PDF file)?

Reviewer #1: Yes

Reviewer #2: No

4. Is the manuscript presented in an intelligible fashion and written in standard English?

Reviewer #1: Yes

Reviewer #2: Yes

5. Review Comments to the Author

Reviewer #1: The authors provide a very well written economic evaluation of a randomized controlled trial examining pulse oximetry and integrated managemetn of childhood illness for diagnosing severe pneumonia. While there are a few places the manuscript could be strengthened, this is as close as I’ve come in recent memory to suggesting a paper for publication without any revisions. I have only minor suggestions to improve the manuscript that should be addressed before acceptance. They are as follows:

- Could use some copy editing to tighten up the grammar and English in a few places, but overall very clearly and thoughtfully written manuscript.

- A clarification is necessary on the final row of Table 1. Is transportation per kilometer correct, if applied as a unit cost? Does this assume that the average unit cost of transport per patient is only for 1 km?

- Page 7 line 141. The authors mention an interest rate on the pulse oximeter. Do they mean a discount rate? If so, should provide justification for the interest rate of 7% when 3% is the industry standard. Otherwise, please explain the need for an interest rate… this is unclear.

- Page 7 line 149. Are all relevant health workers for this kind of intervention mid-level health workers? Is there a possible upper range of salary reimbursement for some medical professionals that is perhaps not captured in the analysis? Would be good to clarify in the text.

- Figures need to be properly labeled. Not all figures have a full description in the text (see Fig2, especially -not all readers will be familiar with a Tornado diagram).

- Lines 329 - 333 (p16-7) It does seem to be a problem that the facility space itself wasn’t formally added to the cost estimate (even if the facilities already exist, some rental costs, utilities, or other amortized capital costs for the space should be provided to the overall estimate - its not correct to say that just because a new building doesn’t need to be built that the use of the space doesn’t incurr cost). In this case, omitting a cost category of this nature would likely lead to an overestimate of the cost effectivness of either program, but have little to no change in the ICER. That said, this seems like a minimal concern because the facilities already exist, and the program is imbedded in large multiuse facilities. For this reason, space is unlikely to be even a moderate cost driver. Tacit recognition of the absence of these costs is sufficient, but the justification and likely impact on the cost effectivness estimates needs to be reworded.

A few additional unnecessary comments to address, but ones you may find worthwhile to consider:

- I understand the conversation around not including ‘final’ health measures. There could be a back-of-the-envelope calcualtion you could give to estimate the number of cases of severe pneumonia that lead to child mortality, though this is unnecessary. In either case, I don’t think this is enough to overtake the underestimate in cost effectivnesss you face by having excluded opportunity costs of care from the patient perspective however, so I dont see this as a substantial limitation of the paper.

- Given the paucity of good primary cost data on these kinds of studies, unless you are planning a separate costing exercise, you should consider a table breakdown of the average cost at each of the facilities in the study, and provide some characteristics of these facilities to help explain how costs differ by study setting. This would be a major value add to the field and would only strengthen the overall publication.

- Providing any additional detail of the sample of participants out of pocket costs would also be helpful in framing the relevant population under study. This data can probably be cross referenced from the RCT, but a recitation of those characteristics here would add value.

Reviewer #2: This paper has the potential to make an important contribution to the literature. The authors do define a specific objective to compare the cost-effectiveness of combined use of pulse oximetry with standard IMCI guideline compared with the standard IMCI guidelines alone to improve diagnostic precision of severe childhood pneumonia in rural Ethiopia. In the methods section, they narrow the scope a bit by focusing on opportunity costs of combined use of Pulse oximetry and the standard IMCI guideline versus the standard IMCI guideline in an Ethiopian rural setting. This paper would benefit from a careful explanation of the methods using best practice and guidance from the literature on CEA of diagnostic approaches. As written, I can’t really assess whether what the authors have done is a cost-consequence analysis or truly a cost-effectiveness analysis. It may be useful to be more explicit and systematic about the methods throughout the paper. For example, justify your choice of effectiveness, excluding all benefits of improved dx in intro, be clear in methods why some treatment costs included, but averted treatment costs not included, and add any caveats to this approach in discussion). It may be the authors want to focus on cost-effectiveness of an approach to detect pneumonia, rather than cost effectiveness of diagnosis, which has implications for the analysis. For instance, CEA of diagnostic tests/approaches are stronger when looking at cases ‘accurately’ diagnosed. This information is not available. In addition, the authors do not explicitly look at a health outcome per se, which is also a hallmark of cost effectiveness analyses, if not looking at diagnostic accuracy. I understand the authors saw this as a caveat, but it is worth including these issues in the methods, rather than a caveat.

Introduction

1. Line 55-57: Feasibility of adding a new technology that requires training in a situation where even a more basic established protocol (IMCI guidelines) can’t be scaled….Pulse oximeter is a medical supply, requires training, maintenance…Implementation of standard IMCI is difficult due to lack of trained staff, lack of drugs and supplies and inconsistent supervision. Can the authors explain how adding additional diagnostic tools and supplies (for oxygen therapy) at added cost addresses/overcomes the health system constraints described and the consequent IMCI ‘know-do’ gap? The justification for this could be stronger in the intro (and the authors may want to revisit this in the discussion).

Methods

1. Line 115, Page 5- the health effect is the number of severe pneumonia cases detected—do the authors have any information cases were correctly detected? (from looking at the RCT results, I believe not). Or, if the cases detected were treated with pre-referral drugs, or referred to a primary hospital for treatment? From table 1, it appears patients in both settings did receive oxygen treatment, drugs and IV fluids. Would it be possible to look at improvements in dx and treatment at the health center level?

2. Intervention costs from provider and patient perspective—

a. The authors capture treatment costs; however, they do this by capturing the patient out-of-pocket expenses, which include treatment costs (drugs and supplies). Do the authors consider this the full opportunity costs to the patient? Or do these costs also represent the health system costs of treatment at the health center level, since it captures drugs, supplies and a government surcharge of 25% surcharge, which presumably captures personnel and overhead costs. Suggest the authors be more explicit about this in methods, and revise, line 133 to read, “We therefore included treatment costs as part of patient costs (Table 10).”

i. If these treatment costs represent both patient cost and health system costs, you may want to justify by indicating you captured treatment costs on the patient side to avoid double counting.

ii. Are these intended to represent opportunity costs to the patient and health system, or just to patient? If just patient opportunity costs, please indicate more clearly this is your measure of patient opportunity costs.

b. Are there any averted costs associated with untreated severe pneumonia, such as hospitalization or longer hospitalization stays? If yes, and these are not captured in this analysis, suggest being explicit that this analysis does not include averted health system and patient costs from improved dx.

c. Related to averted costs due to improved dx, it does appear that this analysis does not look at treatment outcomes and net costs (intervention costs plus incurred health system treatment costs minus health system and patient medical costs averted), however, (1) it is a little confusing why the authors are capturing the treatment (partial? Pre-referral?) costs to the patient at the health center level; and (2) why they don’t include full range of benefits from improved dx. It would be useful for the authors to clarify this in intro, methods, and/or explicitly note this in the discussion.

d. Page 7. Line 151. First indicate how total cost was calculated. This isn’t clear. You estimate costs for each arm (intervention and control) using input costs shown in table 1, but how do you get a total cost for each arm, and then describe the three three average costs (all pneumonia, severe, non-severe)?

e. Page 9, table 2—how did the authors pick the min and max estimates for costs?

3. Cost-effectiveness model

a. The authors develop a model that compares the opportunity cost of combined use of Pulse oximetry and the standard IMCI guideline versus the standard IMCI guideline in an Ethiopian rural setting. I find the wording of this strange, because they are comparing approaches on an intermediate outcome, i.e. the measure of effectiveness is the proportion of children diagnosed with severe pneumonia in the intervention and control. Perhaps just refine the statement on what they are comparing and be more explicit up front about comparing opportunity cost and intermediate outcomes.

b. A cost effectiveness analysis ought to evaluate what occurs after the diagnosis, and fully evaluate the health and cost outcomes for each diagnosis. This could be an improvement in diagnostic accuracy, or changes in health outcomes, resulting from diagnosis and treatment.

i. The authors should explain why additional benefits (i.e. improved health outcomes and averted costs to the health system and patient—averted medical costs, averted health system expenditures), from improved dx, were not included.

ii. Can the authors justify why they chose the measure of effectiveness they did?

c. To the points above, I recommend that the authors more clearly describe their cost-effectiveness model and better define the incremental cost effectiveness ratio up front, and not leave it to the discussion section and the caveat. This is an important issue. I don’t disagree with what the authors say, but one may also wonder, if pneumonia is a well-established leading cause of death and they have information on numbers treated and their costs, then why not model that out? Explain up front, why this approach was not done, or feasible. And to support your approach, reference Drummond et al in the methods section.

i. Perhaps the authors can review this useful check list to justify their approach. Kip, Michelle MA, et al. "Toward alignment in the reporting of economic evaluations of diagnostic tests and biomarkers: the AGREEDT checklist." Medical decision making 38.7 (2018): 778-788.

d. Table 2. Input parameters. Do the costs of diagnosed severe and non-severe pneumonia include the cost of treatment at the health center? If so, shouldn’t the input parameter read, cost of dx and pre-referral treatment of severe pneumonia…’ if and when that is the case?

e. How are Table 2 and 3 related? Is Table 3 JUST diagnostic costs, without treatment? If treatment costs aren’t used in the ICER, I’m not sure why they are included in the cost analysis.

f. Pg. 10, Line 198-199- What was the time horizon for the actual RCT study? The total and average costs have been calculated from a sample over a given period (i.e. one year or two years?)- I believe this is the correct time horizon, if they are calculating total costs and dividing by all diagnosed cases in intervention and control.

Results and Discussion

1. Since effectiveness estimates are the result of another study (referenced), perhaps just refer back to Table 2, and do not make this a result for this study. It is an input into your model.

2. Could you add a new table to show total costs, numbers of cases and average costs for (1) Intervention dx severe (2) intervention non-severe; (3) Control dx severe, and (4) control non-severe –so four columns, and this would correspond to the input costs shown in Table 2, but the reader can understand better how they were derived. You could include information from previous RCT study shown in lines 205 and 206, p. 10 in this table, as it is a given input into your analysis, not a result.

3. I don’t understand the relationship between Table 1 and 3. If in methods you describe how you calculate total cost, this may become clearer.

a. I would think Table 3 would present a summary and disaggregated cost categories for total costs, corresponding to lines 212 and 213, but these look like average unit costs again. It makes more sense to me to look at cost shares of the total costs disaggregated by cost categories.

4. Table 4 and CE results seem more like a cost analysis to me. In the discussion, the authors note that comparisons are difficult to make, in part b/c other similar studies do extend the analysis out to cases treated or DALYs averted. I think this paper could be reframed as a cost consequence analysis and would be an excellent complement to the published RCT. Especially since much of the discussion focuses on costs.

Minor

1. Line 141, what are ‘initial costs’ for pulse oximeter?

2 Add year of currency to Table 1

3. Add currency and year to Table 3 and indicate these are costs for diagnosing severe pneumonia.

4 Since the approach is incremental to the existing IMCI health program, I think the authors can omit lines 330 to 333—unless the improvements used more space/bldg., this would net out to be zero.

6. PLOS authors have the option to publish the peer review history of their article (what does this mean?). If published, this will include your full peer review and any attached files.

**Do you want your identity to be public for this peer review?** For information about this choice, including consent withdrawal, please see our Privacy Policy.

Reviewer #1: No

Reviewer #2: No

---

## [Decision Letter · Decision Letter 1]

7 Jul 2022

Cost-effectiveness of pulse oximetry and integrated management of childhood illness for diagnosing severe pneumonia

PGPH-D-22-00195R1

Dear Mr Tesfaye,

We are pleased to inform you that your manuscript 'Cost-effectiveness of pulse oximetry and integrated management of childhood illness for diagnosing severe pneumonia' has been provisionally accepted for publication in PLOS Global Public Health.

Best regards,

Melissa Morgan Medvedev, M.D., Ph.D.

Academic Editor

Reviewer Comments (if any, and for reference):

Reviewer's Responses to Questions

**Comments to the Author**

1. If the authors have adequately addressed your comments raised in a previous round of review and you feel that this manuscript is now acceptable for publication, you may indicate that here to bypass the “Comments to the Author” section, enter your conflict of interest statement in the “Confidential to Editor” section, and submit your "Accept" recommendation.

Reviewer #1: All comments have been addressed

2. Does this manuscript meet PLOS Global Public Health’s publication criteria? Is the manuscript technically sound, and do the data support the conclusions? The manuscript must describe methodologically and ethically rigorous research with conclusions that are appropriately drawn based on the data presented.

Reviewer #1: Yes

3. Has the statistical analysis been performed appropriately and rigorously?

Reviewer #1: Yes

4. Have the authors made all data underlying the findings in their manuscript fully available (please refer to the Data Availability Statement at the start of the manuscript PDF file)?

Reviewer #1: Yes

5. Is the manuscript presented in an intelligible fashion and written in standard English?

Reviewer #1: Yes

6. Review Comments to the Author

Reviewer #1: No additional comments.

7. PLOS authors have the option to publish the peer review history of their article (what does this mean?). If published, this will include your full peer review and any attached files.

**Do you want your identity to be public for this peer review?** For information about this choice, including consent withdrawal, please see our Privacy Policy.

Reviewer #1: No
